# Developing a simple, cost-effective proof-of-concept vaccine candidate against EHEC for cattle

Cecilia M. Duarte[1,2☉], Laura A. Basile[1,2☉], Diego G. Noseda[1,2], Pierina Fasolo[2], Pilar Leguiza ![ORCID][1,2], Mara S. Roset[1,2*], Gabriel Briones ![ORCID][1,2*]

1 Instituto de Investigaciones Biotecnológicas, Universidad Nacional de San Martín (UNSAM), Consejo Nacional de Investigaciones Científicas y Técnicas (CONICET), Argentina, 2 Escuela de Bio y Nanotecnologías (EByN), Universidad Nacional de San Martín, Buenos Aires, Argentina

☉ Both authors contributed equally to this work.

* gbriones@iib.unsam.edu.ar (GB); mroset@iib.unsam.edu.ar (MSR)

## Abstract

Enterohemorrhagic *Escherichia coli* (EHEC) is a zoonotic pathogen responsible for severe human diseases, including hemorrhagic diarrhea and hemolytic uremic syndrome (HUS), primarily mediated by Shiga toxins (Stx). Intestinal colonization depends on the type III secretion system (T3SS), which induces attaching and effacing (A/E) lesions. Cattle are the main reservoir, and preharvest vaccines are key to reducing human exposure. In this work, we engineered a chimeric antigen EIT (EspA$^{36-192}$-Intimin$^{653-935}$-Tir$^{258-361}$) to be expressed in the periplasm of *E. coli* BL21, enabling simple extraction by thermal shock. To favor scalable production, the process avoids the use of antibiotics, chemical inducers, and mechanical disruption, while remaining compatible with standard infrastructure and low-cost adjuvants. Immunization of mice with an EIT-enriched periplasmic fraction (EIT-PF) induced strong antibody responses with enhanced functional activity against A/E pedestal formation *in vitro*, and accelerated clearance of experimental *E. coli* O157 infection in mice. A three-dose EIT-PF immunization was tested in cattle. Notably, a single EIT-PF dose elicited antibodies that recognized multiple EHEC serotypes and EPEC, as well as each chimera component, effectively interfering with T3SS-dependent pedestal formation *in vitro*. Our findings support EIT-PF as a broadly reactive, proof-of-concept vaccine candidate for cattle; further studies are required to assess efficacy under field conditions.

## Introduction

Since its first isolation by Theodor Escherich in 1884, *Escherichia coli* has become one of the most extensively studied microorganisms and a cornerstone in multiple fields, ranging from basic biology to biotechnology [1]. It has been described that

**Data availability statement:** All raw data and uncropped images underlying the findings of this study are available in the Figshare repository at: DOI: 10.6084/m9.figshare.30764966. All processed images and additional analyses are provided within the manuscript and its Supporting Information files.

**Funding:** This study was financially supported by the Agencia Nacional de Promoción Científica y Tecnológica (ANPCyT, Argentina) in the form of a grant (PICT-2021-CAT-I-00049) received by GB. This study was also financially supported by the Consejo Nacional de Investigaciones Científicas y Técnicas (CONICET, Argentina) in the form of grants received by MSR (PIP 2021–2023 No. 11220200102517CO) and GB (P-UE No. 0086). The funders had no role in study design, data collection and analysis, decision to publish, or preparation of the manuscript.".

**Competing interests:** The authors have declared that no competing interests exist.

colonizing the gut by *E. coli* is a critical step towards normal and healthy intestinal functioning [1]. Although the majority of bacteria in the normal intestinal microbiota are obligate anaerobes, *E. coli* stands out as the most prominent facultative anaerobic commensal inhabiting the intestinal mucosal layer [2].

In addition to being a bacterial commensal, *E. coli* includes pathogenic variants, or pathotypes, such as enteropathogenic *E. coli* (EPEC) and enterohemorrhagic *E. coli* (EHEC) [3]. Both strains share a critical virulence trait: the type III secretion system (T3SS), encoded in the LEE pathogenicity island, which shows strain-specific variations and is thought to have been acquired through horizontal transfer [4]. This T3SS functions as a protein nanomachine that injects effector proteins into host cells, disrupting the cytoskeleton to cause microvilli effacement and actin pedestal formation [5,6], and consequently these strains have been classified as attaching-effacing *E. coli* (AEEC). Critical effectors include Tir, which is translocated into the host cell via the T3SS and inserted into the plasma membrane, where its extracellular domain binds the bacterial adhesin Intimin, initiating signaling cascades that drive actin polymerization and pedestal formation. EspA, in turn, assembles into filamentous structures that form the T3SS translocation channel, enabling intimate bacterial attachment and efficient effector delivery [7,8].

In contrast to EPEC, EHEC produces Shiga toxin (Stx), which is responsible for its distinctive clinical manifestations [9]. Not all Shiga toxin-producing *E. coli* (STEC) strains are associated with human disease; however, enterohemorrhagic *E. coli* (EHEC) represents a clinically relevant subset capable of causing severe complications [5], with *E. coli* O157 being the most prevalent serotype [10]. EHEC strains have been reported as a major cause of foodborne outbreaks [11,12], often leading to bloody diarrhea and hemolytic uremic syndrome (HUS), which involves renal failure, anemia, and low platelet count. Importantly, efficient Stx delivery depends on intimate EHEC adherence to intestinal epithelial cells, a process mediated by the formation of characteristic attaching and effacing (A/E) lesions and actin pedestals [10,11]. Cattle are the primary reservoir of EHEC, with an estimated 5–6% of the global population colonized, and bovine-derived foods representing a major route of transmission [10,13]. Additionally, raw vegetables can carry EHEC due to contamination with cow manure. These findings underscore the importance of controlling EHEC in cattle to reduce human infections [12]. Several strategies have been proposed. Probiotic approaches aim to modulate cattle microbiota and competitively suppress pathogens [14]. Bacteriophages, with their strain-specific action, offer targeted elimination of EHEC without disturbing the overall microbial balance [14]. Pre-harvest vaccination represents a key component of integrated strategies to control EHEC colonization [15]. Econiche, a vaccine based on culture supernatants of pathogenic *E. coli* O157 containing virulence-associated proteins (EspA, EspB, EspD, Tir, Intimin), demonstrated efficacy by reducing intestinal colonization, decreasing the proportion of animals shedding the pathogen, and limiting both the duration and magnitude of shedding [16]. Despite these results, Econiche production was discontinued and is no longer available [17]. Adoption of a vaccination program by cattle producers relies not only on vaccine efficacy but also on costs, which can be prohibitive, particularly

for small-scale operations. Effective and affordable vaccines are therefore essential to ensure widespread implementation [18]. In this context, the withdrawal of Econiche underscores the need for a simple, sustainable, and low-cost pre-harvest vaccine against EHEC.

In Argentina, infection with Shiga toxin–producing *Escherichia coli* (STEC) and the associated hemolytic uremic syndrome (HUS) remains an endemic public health problem, contrasting with the more epidemic pattern observed in other regions where outbreaks are sporadic. National surveillance continues to report one of the highest global incidences of HUS in children under five, with STEC O157:H7 and non-O157 serotypes such as O145:H28 frequently implicated [19,20]. Recent multicenter surveillance confirmed that approximately 4% of pediatric bloody diarrhea cases were STEC-positive, and a considerable fraction progressed to HUS, underscoring the persistent burden of disease [21]. Furthermore, studies in cattle, the main reservoir of STEC, have documented sustained prevalence rates at slaughterhouses, reinforcing the role of livestock in maintaining transmission [22]. This ongoing endemic scenario highlights the need for preventive interventions targeting the animal reservoir, including vaccines designed to reduce bacterial colonization and limit human exposure.

Here, we designed a chimeric antigen EIT (EspA$^{36-192}$-Intimin$^{653-935}$-Tir$^{258-361}$), combining immunogenic domains of key virulence factors [23]. The protective efficacy of EIT has been previously demonstrated when expressed in non-virulent bacterial carriers, including *Brucella abortus*, *Salmonella enterica*, and *Lactobacillus acidophilus*, thereby supporting its validity as a vaccine antigen against EHEC in murine models [24–26]. In this study, we explore a simple and cost-effective production strategy by expressing EIT in *Escherichia coli* BL21, a biosafe laboratory strain commonly used for recombinant protein production. Following validation in mice, the immunogenicity of this formulation was preliminarily evaluated in steers using an initial protocol designed for safe and scalable production.

## Materials and methods

### Bacterial strains and growth conditions

The recombinant antigen EIT was expressed in *E. coli* BL21 (DE3). Challenge infections in mice were performed with EHEC O157:H7 (EDL933), and actin pedestal-inhibition assays employed EPEC O127:H6 (E2348/69). All EHEC/EPEC work was conducted under BSL-2 conditions. Cultures were maintained at 37 °C in LB medium, with antibiotics added when necessary: ampicillin 50 µg/ml, kanamycin 50 µg/ml, and nalidixic acid 20 µg/ml.

### Design of the EIT antigen

The recombinant EIT antigen was derived from plasmid pTRC-EITH7 [25]. A SalI fragment containing an A-rich region, the Trc promoter, and the chimeric EspA$^{36-192}$–Intimin$^{653-935}$–Tir$^{258-361}$ (EDL933) was ligated into the pBBR1MCS-2 vector (Km$^r$) and transformed into *E. coli* BL21 (DE3) competent cells. A (EAAAK)$_4$ spacer was inserted between antigen domains to promote proper folding, and a β-lactamase signal sequence (β-lac$^{1-35}$) was added at the N-terminus to target the chimera to the periplasm [25]. The resulting plasmid, pTRC-EIT-K, was confirmed by PCR and DNA sequencing (Macrogen Inc.).

### Production of the EIT recombinant protein

A starter culture of *E. coli* BL21 carrying pTRC-EIT-K was grown overnight and used to inoculate 800 mL of LB broth (1:20 dilution). EIT expression was induced with an autoinduction mixture [27]: glycerol 5 g/L, glucose 0.5 g/L, and lactose 2 g/L, avoiding IPTG. Cultures were incubated for 6 h at 37 °C with shaking at 200 rpm (OD600~2.5), then harvested by centrifugation and stored at −20 °C. Periplasmic proteins were initially released using osmotic shock [28]; however, due to the complexity of this method, a thermal shock approach was adopted for scalability [29]. Briefly, the bacterial pellet was resuspended in 2% of the original culture volume in 50 mM Tris-HCl (pH 7.6) containing 10 mM EDTA, incubated on ice for 20 min, and then heated at 55 °C for 40 min. This procedure resulted in an estimated 50-fold concentration of the

periplasmic fraction. After centrifugation, supernatants containing the periplasmic fraction were collected and stored at −20 °C until use. Bacterial survival during thermal treatment was assessed by collecting samples every 10 min, plating appropriate dilutions on LB agar, and counting CFUs. *E. coli* BL21 carrying the empty pBBR1MCS-2 vector served as a control in all experiments.

Recombinant EIT recovery was monitored by Western blot analysis in the Odyssey Imaging System (LI-COR), using α-EITH7 mouse primary antibodies and α-mouse IRDye fluorophore-labeled secondary antibodies (LI-COR) as previously described [26]. Total protein content was quantified by the Bradford method. EIT was quantified in Western blot by performing a standard curve with purified recombinant MBP-EITH7, with the assistance of ImageJ software (Integrated-Density measures).

## Plasmid stability in the absence of antibiotic pressure

To evaluate if EIT might be produced in the absence of antibiotics, an analysis of plasmid stability was carried out. *E. coli* carrying the pTRC-EIT-K plasmid was cultured overnight in LB-kanamycin. Three successive subcultures at a dilution ratio of 1:1,000 in LB media without antibiotics were grown overnight. The number of generations (g) was calculated as $g = \log_2 (N_f / N_0)$, where $N_f$ and $N_0$ are the final and initial bacterial numbers, respectively, and were estimated from $OD_{600}$ determinations. Aliquots of each subculture were diluted and spread on LB agar plates with and without antibiotics. Plasmid stability was determined through CFU counting in the presence or absence of kanamycin. Significant differences were analyzed by one-tailed Student´s *t*-tests.

## Ethics statement

All animal studies were reviewed and approved by the Committee on the Ethics of Animal Experiments of the Universidad Nacional de San Martín (UNSAM) (permit number CICUAE UNSAM 15/2018), in accordance with international guidelines (Helsinki Declaration and amendments, Amsterdam Protocol on animal welfare, and NIH Guide for the Care and Use of Laboratory Animals). Mice were monitored daily for signs of stress or discomfort and were euthanized by $CO_2$ inhalation at the end of the study, following institutional guidelines for humane endpoints. Cattle were immunized according to the vaccination schedule, and blood samples were collected at defined time points by trained personnel, with all efforts made to minimize stress or discomfort. No additional invasive procedures or euthanasia were performed in cattle.

## Vaccination experiments

**Mouse assays.** Eight-week-old BALB/c mice were randomly divided into four groups of five mice each and ear-tagged for individual identification. Each group was intraperitoneally vaccinated with either 100 µg of EIT-enriched BL21 periplasmic fraction (EIT-PF) with aluminum-based adjuvant (Group A) or EIT-PF without aluminum-based adjuvant (Group B). As controls, 100 µg of periplasmic proteins from BL21 (pBBR1-MCS2) (C-PF) with adjuvant (Group C) and PBS (Group D) were used. Doses were administered at 0, 2, and 4 weeks. Blood was collected one week after the second and third immunizations, and sera were stored at −20 °C until use.

**Steer assays.** The recombinant antigen EIT was evaluated in steers from the INTECH-Chascomús experimental field. Steers aged 18–20 months, weighing approximately 240 kg, were identified as 1–6 (EIT-PF) and 7 (C-PF). Pre-immune sera were collected from all animals. The vaccine dose contained 500 µg of total proteins (EIT-PF or C-PF) and 600 µL of alumina adjuvant (ALUPHARMA) in a final volume of 2 mL (in PBS). Doses were administered subcutaneously at the neck at 0, 2 and 4 weeks. Blood samples (10 mL) were collected before the start of each immunization and finally at week 6. Sera were stored at −20 °C until use.

**Challenge of vaccinated mice and fecal shedding.** Eight weeks after the first immunization, mice were orally challenged with 1 x 10⁹ CFU of EHEC O157:H7 (strain EDL933). The assay was performed in the EIT-PF + adjuvant

vaccinated group. C-PF vaccinated mice were used as control group. Mice were inspected for signs of infection, such as ruffled fur, hunched posture, and tremors, and their feces were collected daily, resuspended in PBS, and plated on MacConkey-Sorbitol agar with nalidixic acid. EHEC was determined by CFU counting. The time elapsed from the start of fecal shedding assay until no EHEC was detected, was used to create Kaplan-Meier survival curves. Statistical differences between curves were analyzed by Log-rank (Mantel-Cox) test in GraphPad Prism software.

## Serum antibody determination by ELISA

Indirect ELISA was used to detect α-EIT-specific IgG antibodies in sera from vaccinated mice and steers. Purified MBP-EITH7 [26] (125 ng/well in sodium carbonate buffer, pH 9.6) was used to coat 96 well-microplates. Briefly, it corresponds to the EITH7 chimeric antigen fused to a maltose binding protein (MBP) in pLC3 vector, expressed in *E. coli* and purified by amylose affinity chromatography. Sera was evaluated at 1:200 or 1:300 dilution in blocking buffer (PBS with 0.10% v/v Tween-20 and 5% w/v dry milk) for mice and steers, respectively. Microplates were incubated with peroxidase-conjugated α-mouse-total IgG (dilution 1:1000 in PBS-T with 1% dry milk) or α-bovine-total IgG (1:4000), and color was developed by the addition of the peroxidase substrate TMB (tetramethylbenzidine, Sigma), according to established protocols. OD at 450 nm was determined. Sera samples were evaluated in duplicate in two independent experiments, and data were analyzed by One-way ANOVA and Tukey contrasts. For antibody titer determination in steers, two-fold serial dilutions of sera (starting at 1:150, in blocking buffer) were performed. Titers were determined using the reciprocal of the highest dilution that resulted in an absorbance value greater than the mean + 3 standard deviations (SD) of the absorbance value from pre-immune sera. Titers were measured for individual steers at each post-immunization point; assays for the control steer (C-PF) were performed in triplicate. Results were reported as the base 2 logarithm of the mean reciprocal endpoint titers. Data were analyzed by One-way ANOVA and Sidak's pairwise comparisons.

To assess pathogen-specific IgG reactivity in steer sera, lysates of EPEC O127:H6 and different EHEC serotypes (O157:H7, O145, O121:H9, O26) were prepared according to [30]. *E. coli* BL21 was used as control strain. Briefly, overnight cultures were diluted 1:20 in DMEM and incubated statically for 4 h at 37° C to induce expression of T3SS proteins. Cultures were pelleted and resuspended in 1 mL PBS. Bacteria were inactivated and lysed by incubation at 65 °C for 1.5 h. Protein concentration was determined by Bradford assay. Lysates were diluted in coating buffer (pH 9.6) and used to coat microplates (1 μg/well). ELISA was performed as described above, using a pool of sera of EIT-PF immunized steers (1:300 dilution) collected at week 2 p.i. (single dose), in triplicate. Data were analyzed by One-way ANOVA and Dunnett contrasts comparing to BL21 strain.

To assess IgG reactivity to individual components of the chimeric antigen in steer sera, purified His-EspA, His-Intimin and His-Tir proteins [24] were quantified by Bradford assay and used to coat microplates (150 ng/well). Briefly, they correspond to the EspA, Intimin or Tir fragments of EITH7 chimeric antigen separately subcloned into pET-26b(+) vector, fused to a His-tag, expressed in *E. coli* and purified by Ni-NTA-His bind Resin columns. Bovine serum albumin (BSA) was used as negative control. ELISA was performed as described above, using a pool of sera from pre-immune or EIT-PF immunized steers collected at week 2 p.i. (single dose), in 1:300 dilution, in triplicate. Data were analyzed by One-way ANOVA and Sidak's pairwise comparisons.

## Inhibition of actin pedestals, staining, and fluorescence microscopy

To evaluate if α-EIT antibodies in the mouse and steer sera were able to neutralize T3SS-dependent actin pedestal formation, we performed an inhibition assay in HeLa cells with EPEC O127:H6. Steer sera were previously inactivated at 56 °C for 30 min. Briefly, HeLa cells were grown in Dulbecco's modified Eagle medium (DMEM, Gibco) supplemented with 5% fetal bovine serum (FBS), seeded onto coverslips placed on 24-well cell culture plates (5 x 10$^4$ cells per well), and cultured for 16–20 h at 37 °C with 5% $CO_2$. For infection assays, an overnight culture of EPEC was diluted 1:25 in DMEM prewarmed at 37 °C to favor T3SS induction and incubated for about 1 h at 37 °C until $OD_{600}$ of 0.5. Each serum

(mouse or steer, 200 µl) was combined with 600 µl of the EPEC culture, and the mix was incubated for 20 min at room temperature. Each mixture was used to infect the HeLa cells on the coverslips in duplicate (400 µl/coverslip). After 1 h incubation, non-attached bacteria were removed, and cells were incubated for an additional 3 h to allow the formation of actin pedestals. HeLa cells were washed and fixed with 4% PFA. For actin staining, cells were first permeabilized with 0.2% Triton-X-100, blocked with 5% BSA, and incubated with Rhodamine Phalloidin (Life Technologies, 1:250 dilution) for 1 h. Finally, coverslips were stained with DAPI (1:1000) for 3 min and visualized by fluorescence microscopy (Nikon 80i). The number of cells containing pedestals to total cells was determined in at least 20 fields for each preparation. Data were analyzed by One-way ANOVA and Tukey contrasts. The detailed step-by-step protocol is available at https://www.proto-cols.io/private/9931ECDCC6E011F0BE490A58A9FEAC02.

## Results

### The chimeric antigen EIT was directed to the periplasm and subsequently released by thermal shock

The chimeric antigen EIT was successfully expressed in *E. coli* BL21(DE3) using the pTRC-EIT-K plasmid, which encodes the C-terminal region of EspA, the Tir-binding domain of Intimin, and the Intimin-binding domain of Tir (Fig 1A). Alpha-Fold modeling predicted a coherent 3D structure for EIT (Fig 1B). To facilitate recovery, EIT was directed to the bacterial periplasm via an N-terminal β-lactamase signal peptide. Periplasmic extraction was achieved using either osmotic shock or thermal permeabilization, with thermal shock providing a higher yield and a simpler protocol (Fig 1C). Full-length EIT

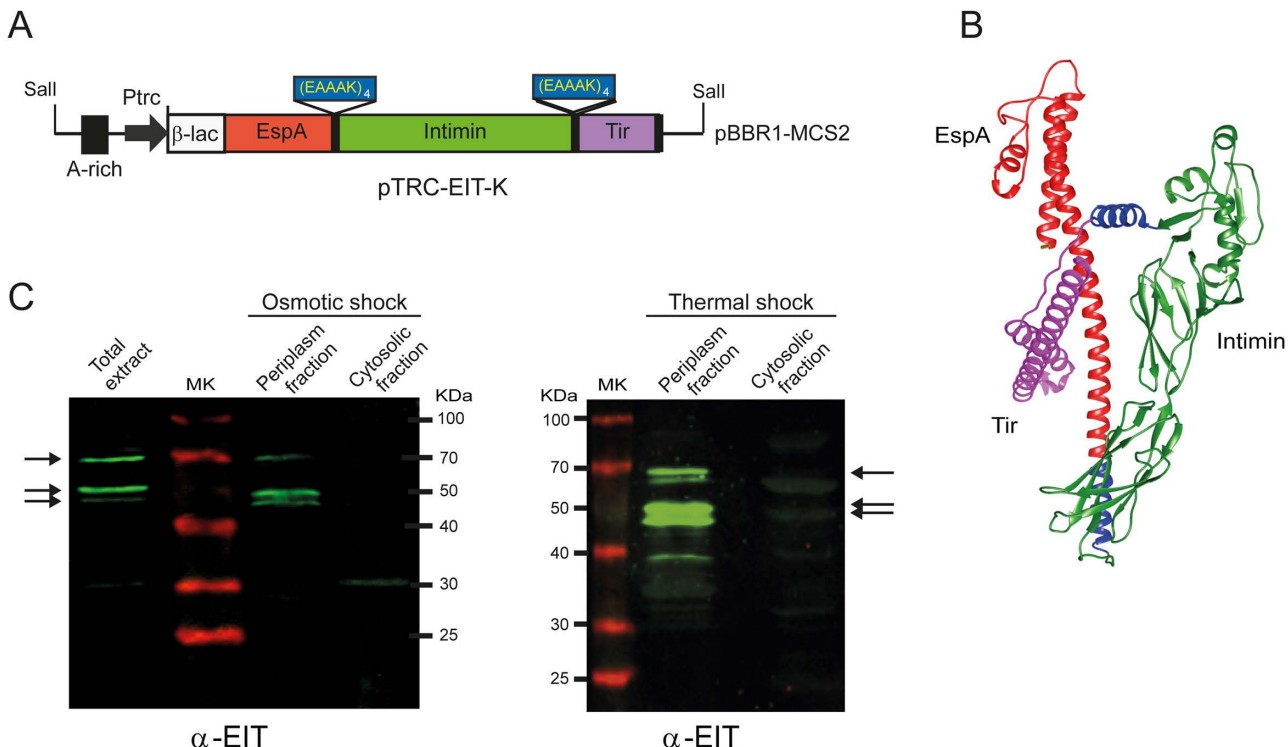

**Fig 1. Design and preparation of the EIT antigen. (A, B)** Sequences encoding the peptides EspA$^{36-192}$ (in red), Intimin$^{653-935}$ (green), and Tir$^{258-361}$ (violet) were fused in-frame, separated by a spacer sequence (blue). **(A)** Schematic representation of EIT construction in pBBR1MCS-2 vector under pTRC promoter control. The β-lactamase signal sequence in the N-terminus directs the chimeric antigen to the bacterial periplasm. **(B)** 3-D structure of the EIT antigen predicted in Alphafold2. **(C)** Western-blot analysis of the recombinant construction in *E. coli* BL21 after an osmotic (left) or thermal (right) shock, using α-EIT primary antibodies.

(~65.1 kDa) and a few smaller antigenic bands (~50 kDa), likely due to proteolytic processing, were detected. Thermal treatment fully inactivated the bacterial culture within 40 min (S1 Fig). Western blot quantification indicated that EIT represented approximately 3% of the total protein in the periplasmic fraction, which was used as the immunogen for subsequent mouse and cattle experiments. All proteins present in this preparation are intrinsic to *E. coli*, ensuring that only bacterial antigens of this bacterium are delivered.

Since EIT expression was plasmid-encoded, and we aimed to avoid antibiotic use during *E. coli* culture and scale-up, we employed pBBR1MCS-2, a stable plasmid that did not require antibiotic selection [31]. To assess plasmid stability in the absence of antibiotic selection, serial 1:1,000 dilutions were performed as described in Materials and Methods. Remarkably, pBBR1MCS-2 remained stable for at least 30 bacterial generations in the absence of kanamycin (S2 Fig).

### EIT-PF immunization elicits a strong immune response in mice

EIT-PF was used to immunize a group of eight-week-old mice in a three-dose scheme, with 100 µg of total protein/dose (Fig 2A). As a control group, mice were immunized with the periplasmic fraction (100 µg/dose) of *E. coli* BL21(DE3) carrying the empty vector pBBR1MCS-2 (C-PF). As shown in Fig 2B, after the second dose (three weeks post-immunization), a significant rise in the level of antibodies against EIT was detected by ELISA when the antigen was combined with the aluminum-based adjuvant (white squares) compared with the rest of the experimental groups: PBS (black circles), C-PF (white circles), and EIT-PF (black squares). Interestingly, at five weeks post-immunization, after the third dose of the EIT antigen, no difference in antibody level was observed between the EIT-PF groups, independently of the addition of the adjuvant (Fig 2B, white and black squares). To confirm antigen specificity, sera from four EIT-PF+Adjuvant immunized mice were evaluated by Western Blot analysis against the recombinant protein MBP-EITH7 (S3 Fig).

### Antibodies induced by EIT-PF immunization efficiently prevented *in vitro* enteropathogenic *Escherichia coli* attachment to host cells and actin pedestal formation

We employed the EPEC pedestal formation assay to evaluate functional antibodies elicited by the chimeric EIT antigen constructed from EHEC EspA, Intimin, and Tir sequences. Both EPEC and EHEC rely on Tir–Intimin interactions to induce actin-rich pedestals. As previously stated, the use of EPEC in these assays is based on the fact that i) EPEC and EHEC share the same T3SS encoded in the pathogenicity island LEE, ii) antigens EspA, Intimin, and Tir are highly homologous in both pathotypes, iii) EPEC is more efficient in actin pedestal formation *in vitro,* and iv) EPEC has no Stx and experiments can be performed in a BLS2 facility [20]. Consequently, the pedestal formation assay using EPEC provides a more sensitive readout for detecting functional antibody activity.

Sera collected from the immunization experiment shown in Fig 2A were evaluated *in vitro* for their ability to inhibit EPEC attachment and pedestal formation at five weeks post-EIT immunization. In control sera-treated cells, actin accumulation dots marking EPEC attachment were clearly visible on HeLa cell surfaces (Fig 3A, left panel), whereas antibodies induced by EIT-PF substantially interfere with EPEC attachment, markedly reducing the number of actin pedestals (Fig 3A, right panel). To quantify the efficiency of the different treatments, the percentage of cells bearing pedestals was determined. Pre-incubation of EPEC with either naïve sera or control periplasmic fraction (C-PF) sera did not inhibit pedestal formation, while sera from animals immunized with EIT, with or without adjuvant, significantly reduced pedestal numbers. Although ELISA results indicated similar antibody levels induced by EIT immunization, the inclusion of an aluminum-based adjuvant substantially enhanced functional activity, improving the antibodies' ability to block pedestal formation (Fig 3B). This enhancement is consistent with the known effects of aluminum adjuvants, which can promote the induction of antibodies with improved functional quality, including higher affinity and neutralizing capacity, beyond simply increasing titers [32]. Based on these findings, the adjuvanted formulation was incorporated in subsequent experiments to achieve stronger functional immunity and more effective pedestal inhibition.

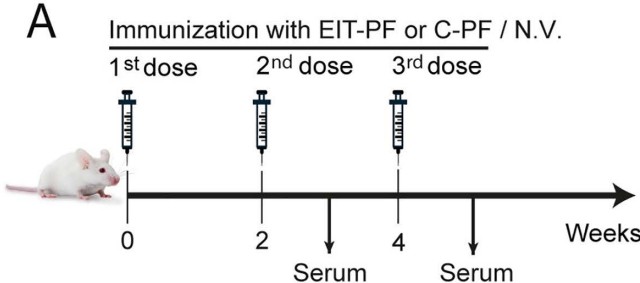

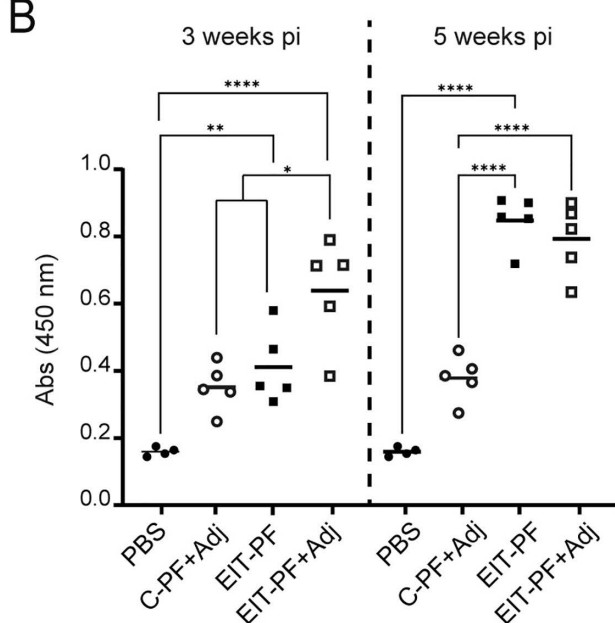

**Fig 2. Mice immunization and ELISA.** (A) Mice were immunized in a three-dose regimen at weeks 0, 2, and 4 by the intraperitoneal route. EIT-PF, C-PF: Protein periplasmic fraction (100 μg) of *E. coli* BL21 culture containing pTRC-EIT-K plasmid (EIT) or an empty control vector (C) after osmotic shock. PBS: PBS-vaccinated mice. Sera were collected at weeks 3 and 5 post-immunization (pi). (B) Indirect-ELISA assays were performed to detect α-EIT-specific antibodies in sera of each group: black circles: PBS, open circles: C-PF plus adjuvant, black squares: EIT-PF, open squares: EIT-PF plus adjuvant. Data were analyzed by One-way ANOVA; asterisks indicate a p-value < 0.05 (*), < 0.01 (**), or < 0.0001 (****) in Tukey contrasts.

### Immunization with EIT-PF elicits an immune response that efficiently controls experimental EHEC infection

A mouse model was adapted to evaluate the protective efficacy of the EIT antigen against experimental infection. BALB/c mice were immunized with the EIT-containing periplasmic fraction following a three-dose schedule (Fig 4A). After completion of immunization, mice were challenged orally with EHEC, and fecal bacterial shedding was monitored over subsequent days. Mice immunized with EIT-PF cleared the infection more rapidly than control animals, with bacterial CFU significantly reduced by day 4 post-challenge (Fig 4B). These findings are consistent with the antibody titers measured by ELISA and the neutralization observed in the pedestal formation assay. Survival analysis further confirmed the protective effect of EIT immunization (Fig 4C; p < 0.01, Log-rank test), with a median clearance time of 4 days for EIT-PF-immunized mice versus 10 days for controls.

A

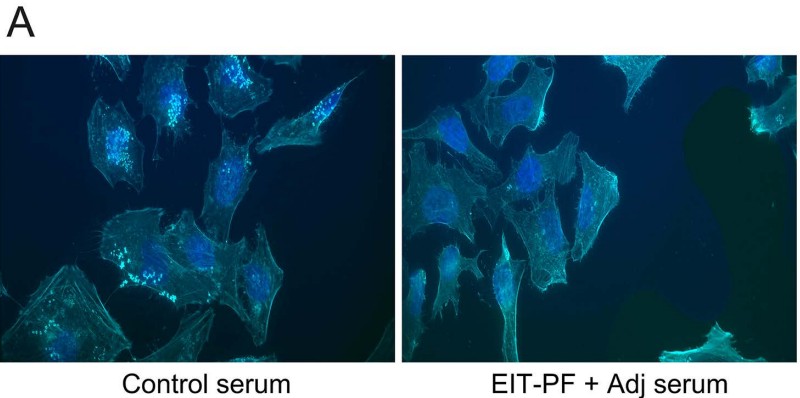

Control serum                    EIT-PF + Adj serum

B

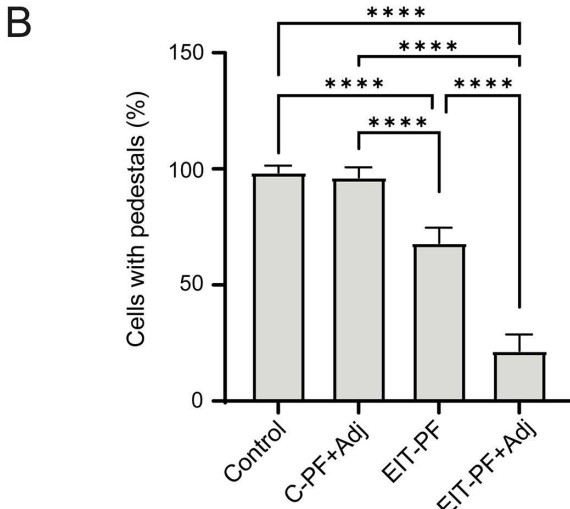

**Fig 3. Formation of actin pedestals in HeLa cells.** The functional capability against EPEC T3SS-dependent actin rearrangements was analyzed in mouse sera of each group. **(A)** Representative microphotographs of EPEC-infected HeLa cells after incubation with control (left) or EIT-PF plus Adjuvant (right) sera. Actin pedestals are evidenced in the left photo by intense, typical dots of Rhodamine Phalloidin mark (in light blue). **(B)** The number of cells carrying pedestals elicited by EPEC, referred to the total cells, was quantified in at least 20 fields for each serum evaluated. Error bars correspond to standard deviations from replicates. Data were analyzed by One-way ANOVA; asterisks (****) indicate a p-value < 0.0001 in Tukey contrasts.

## Cattle immunization with EIT-PF induced a strong antibody response that interferes with EPEC pedestal formation

A group of steers was immunized following a three-dose schedule analogous to that used in mice (Fig 5A). Blood samples were collected prior to the first immunization to obtain pre-immune sera and establish baseline antibody levels. Subsequent sera were collected two weeks after each immunization to assess anti-EIT antibody responses. Notably, a strong antibody response was already observed two weeks after the first EIT-PF dose, while booster doses (second and third) did not produced OD450 signal increases, suggesting that a single dose may suffice to elicit a robust immune response in cattle (Fig 5B). Semi-quantitative analysis of endpoint serum titers further confirmed the antibody responses observed in Fig 5B (S4 Fig). In addition, a pool of sera from one-dose immunized steers was tested for reactivity against multiple EHEC serotypes (O145, O121, and O26, as well as O157) and EPEC O127. IgG antibodies elicited by the EIT antigen specifically recognized lysates from all serotypes cultured under conditions that induce the T3SS (S5 Fig). When the

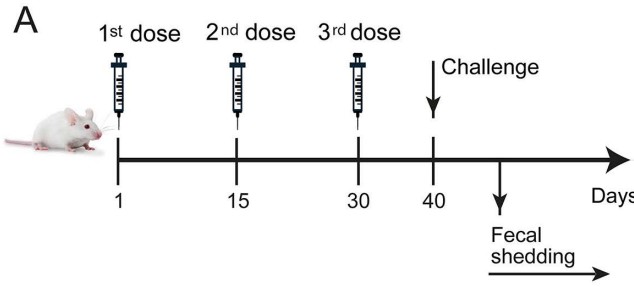

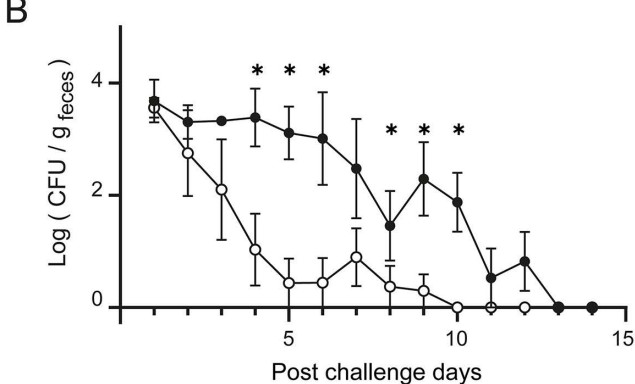

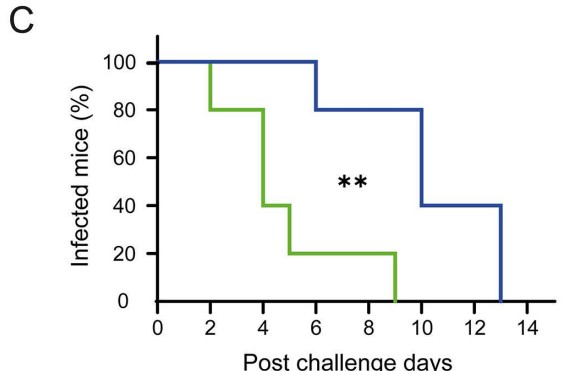

**Fig 4. Challenge of vaccinated mice and fecal shedding assay. (A)** Mice immunized in a three-dose regimen were orally challenged with EHEC. Bacterial fecal shedding was monitored daily by CFU counting. **(B)** EHEC determination on sorbitol-agar plates per gram of feces collected. Open circles: EIT-PF plus Adjuvant vaccinated group, black circles: control (C-PF plus Adjuvant) group. Error bars correspond to standard deviations from biological replicates; asterisks (*) indicate a p-value < 0.05 in Student´s t-tests. **(C)** Percent of mice remaining EHEC-fecal shedding (infected mice) over time, in control (blue) and EIT-PF plus Adjuvant (green) groups, according to Kaplan-Meier survival curves. Asterisks (**) indicate a p-value < 0.01 in log-rank test.

individual components of the EIT chimera (EspA, Intimin, and Tir) were expressed separately, sera from immunized cattle recognized each component in ELISA assays, demonstrating that antibodies elicited by the EIT antigen target all major functional domains of the chimera (S6 Fig). These results confirm that the immune response is directed against multiple epitopes within the chimeric antigen, supporting its potential as a broadly reactive proof-of-concept vaccine candidate.

The functional activity of sera was evaluated using the actin pedestal formation *in vitro* assay. Sera collected at different time points post-immunization were tested for their ability to inhibit EPEC-induced pedestal formation. Representative images illustrate the marked reduction in infected cells when EPEC was pre-incubated with sera from EIT-PF-immunized

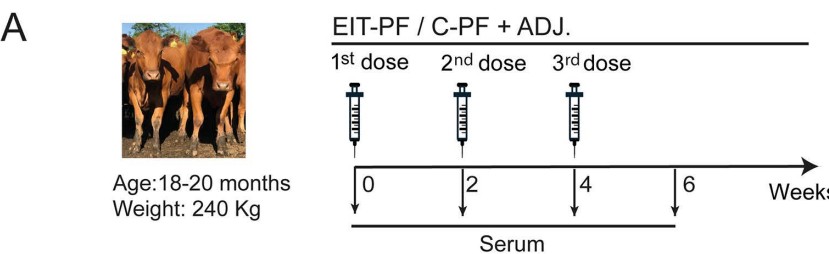

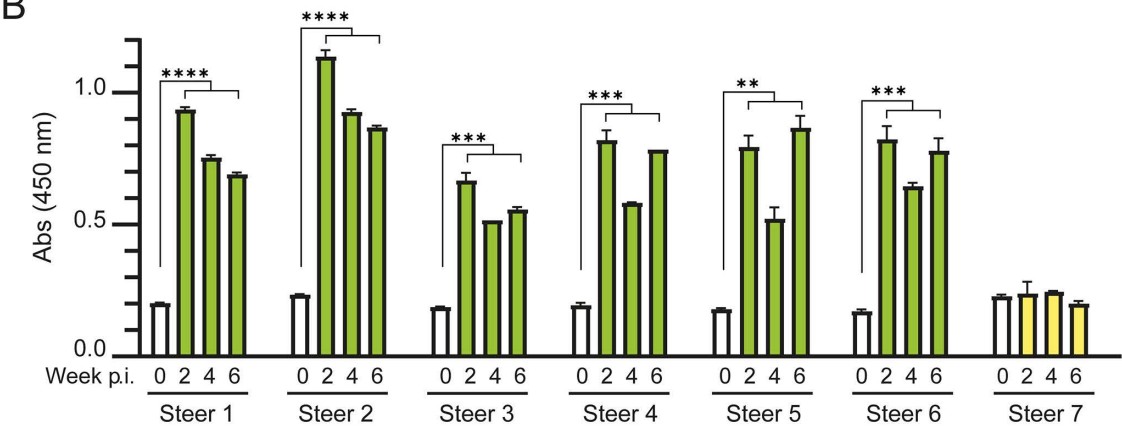

**Fig 5. Steer immunization and ELISA. (A)** Steers were immunized in a three-dose regimen at weeks 0, 2 and 4 by subcutaneous route. EIT-PF, C-PF: Protein periplasmic fraction (500 µg) of *E. coli* BL21 culture containing pTRC-EIT-K plasmid (EIT) or an empty control vector (C) after thermal shock. Sera were collected at 0-, 2-, 4- and 6-weeks post-immunization (pi) **(B)** Indirect-ELISA assays were performed to detect α-EIT specific antibodies in sera of each steer at different pi times. White bars: time 0 (pre-immunization); green bars: EIT-PF plus adjuvant doses, yellow bars: C-PF plus adjuvant doses. Data were analyzed by One-way ANOVA; asterisks indicate a p-value < 0.01 (**), < 0.001 (***) or < 0.0001 (****) in Tukey contrasts.

steers (Fig 6A). Quantitative analysis confirmed that all EIT-PF-immunized steers (1–6) developed antibodies capable of interfering with the type III secretion system (T3SS), significantly impairing pedestal formation even after a single dose (Fig 6B). In contrast, pre-immune sera and sera from a steer immunized with the control periplasmic fraction (C-PF, steer 7) exhibited no functional activity. These results demonstrate that EIT-PF effectively induces antibodies that disrupt T3SS-mediated effector functions and inhibit AEEC adhesion to host cells.

## 4. Discussion

Cattle can harbor bacteria in their digestive systems with zoonotic potential, which can be spread to the environment through their feces. Notably, bacteria such as *E. coli*, *Salmonella*, or *Campylobacter* are the most conspicuous bacterial pathogens found in cattle feces [33]. Transmission to humans can occur through direct contact with animals, with contaminated environments, and by consumption of contaminated meat or dairy products. EHEC colonizes the animal's terminal rectum [34] and as was mentioned, the colonization step requires the attachment to intestinal cells, a process that depends on the T3SS [35].

Although EHEC has often been considered harmless to adult cattle, increasing evidence indicates that it can act as a pathogen, particularly in calves younger than one month [13]. Shiga toxin has been reported to impair the bovine immune system [36], and in newborn calves, EHEC infection has been associated with diarrhea and characteristic attaching-and-effacing (A/E) lesions in both the large and small intestine [36]. Moreover, the prevalence of EHEC in cattle rises under stressful conditions such as long-distance transport, dietary changes, or antibiotic treatment, events that alter the intestinal

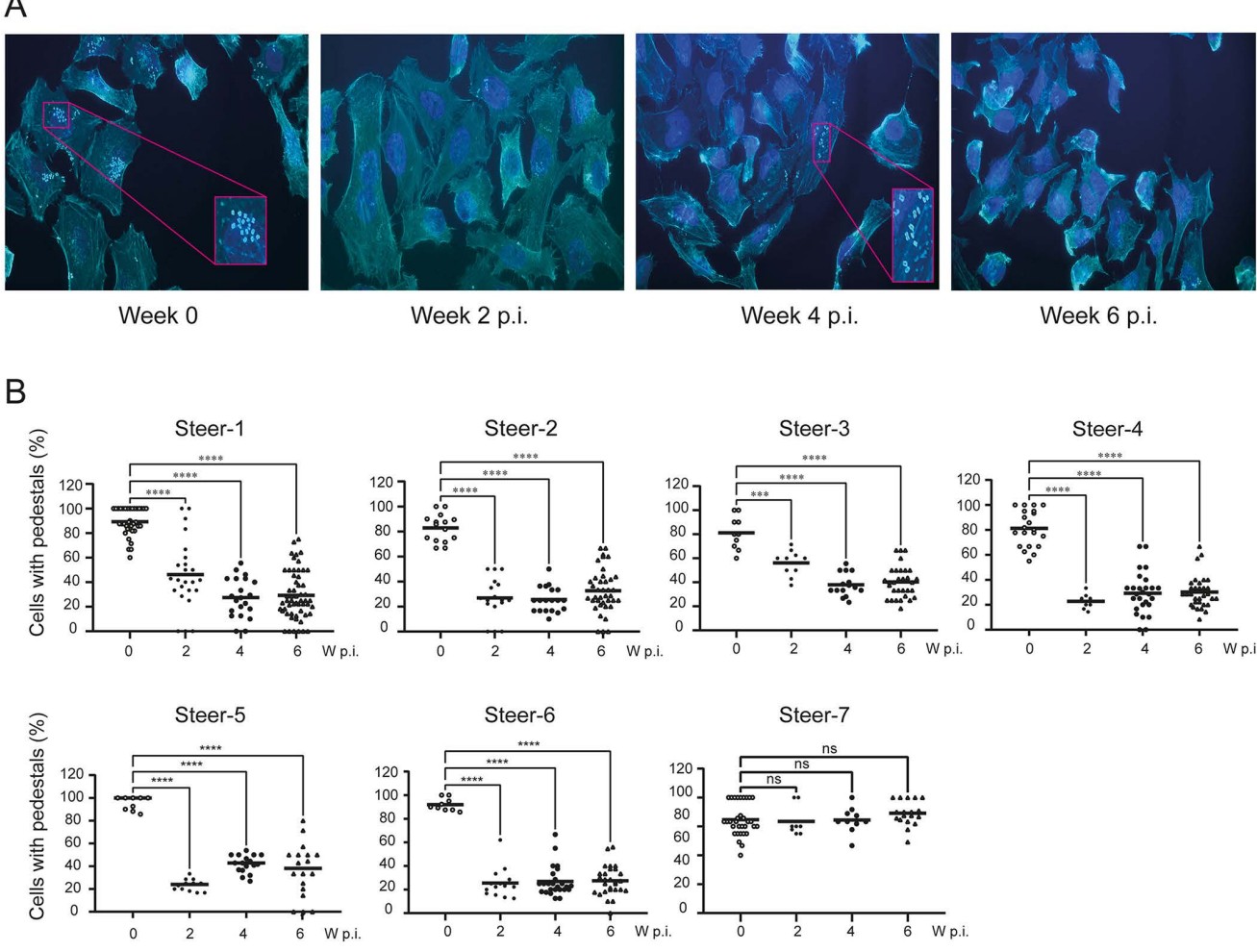

**Fig 6. Formation of actin pedestals in HeLa cells.** The functional capability over EPEC T3SS-dependent actin rearrangements was analyzed in the serum of each steer at each pi time. **(A)** Representative microphotographs of EPEC-infected HeLa cells after incubation with serum from an EIT-PF-vaccinated steer at weeks 0 (pre-immunization serum), 2, 4 and 6 pi. Actin pedestals are evidenced by intense typical shapes of Rhodamine Phalloidin mark (in light blue). Sections showing actin-pedestals were zoomed in. **(B)** The number of cells carrying pedestals, referred to the total cells, was quantified in at least 20 fields for each serum evaluated. Data were analyzed by One-way ANOVA; asterisks indicate a p-value < 0.001 (***) or < 0.0001 (****) in Tukey contrasts. ns: not significant.

microbiota and modulate immune responses [37]. Given its role in zoonotic transmission, preharvest interventions such as vaccination have been proposed to reduce EHEC shedding. Indeed, preharvest vaccines have proven effective in limiting bacterial dissemination into the environment [38]. These approaches are particularly important to prevent EHEC infections in humans, which in children can progress to hemolytic uremic syndrome (HUS) [39,40]. Nevertheless, Econiche, one of the most effective vaccines developed against EHEC colonization in cattle, was discontinued despite clear evidence from field trials demonstrating reduced fecal shedding. Adoption by producers remained limited, highlighting that cost and logistics are major barriers to implementation.

With this in mind, we aimed to develop a simplified and economically sustainable vaccine strategy against EHEC. We reasoned that, in the highly competitive bovine vaccine market, production costs must be minimized to ensure widespread adoption. Beyond selecting optimal antigens, we focused on downstream processes to reduce expenses. First,

to eliminate the need for antibiotic selection, we cloned the EIT antigen ORF into the inherently stable pBBR1MCS-2 backbone [31], generating pTRC-EIT-K. This plasmid remained stable after more than $10^9$-fold dilutions in the absence of kanamycin (S2 Fig), and even during murine infections [31]. Second, to avoid costly IPTG induction, we adopted an autoinduction system in which cultures grown in a multi-sugar medium (glucose, glycerol, lactose) naturally switch from biomass accumulation to lac-promoter–driven expression [27]. This strategy eliminates the need for chemical inducers and reduces carbon-source costs [39]. Moreover, by expressing the antigen in the non-pathogenic, protein-expression– friendly *E. coli* BL21 and targeting it to the periplasm, we enabled its recovery via thermal shock, obtaining an enriched extract capable of eliciting a specific antibody response under standard BSL-1 conditions. Although production costs need to be validated at pilot scale, this workflow highlights key cost-saving steps toward achieving a financially viable preharvest vaccine for cattle [41].

The use of a periplasmic fraction rather than a fully purified antigen is justified on both technical and practical grounds, as it preserves native-like folding, reduces processing steps, and substantially lowers production costs -key considerations for developing a commercially viable cattle vaccine. Because protection depends on antibodies recognizing conformational epitopes involved in the Intimin–Tir interaction, periplasmic extraction helps maintain proper folding relative to more extensive purification procedures [42], thereby increasing the likelihood of eliciting functional, pedestal-inhibiting antibodies. Although the periplasmic fraction naturally contains additional *E. coli* proteins, which are themselves antigenic, our pedestal-neutralization assays demonstrated that periplasmic preparations from BL21 cells lacking the EIT antigen did not confer protective activity. This confirms that the functional immune response is attributable specifically to EIT rather than to background periplasmic components. Moreover, minimizing purification steps shortens the production workflow, facilitates scalability, and aligns with the economic constraints of real-world cattle vaccination programs.

A single immunization with EIT-PF in steers elicited functional antibodies capable of interfering with *in vitro* T3SS-dependent pedestal formation; however, the absence of a booster effect should be interpreted cautiously, potentially reflecting true saturation or variability unresolved with the current sample size. In addition, although pre-immunization sera showed only background reactivity in both ELISA and functional assays—suggesting no ongoing or recent infection—we cannot completely rule out the possibility of previous natural exposure, as transient colonization with EHEC or related *E. coli* strains is common in cattle and may prime the immune system without leaving persistent serological signatures.

The limited number of steers in this exploratory study reflects standard ethical and logistical constraints for initial large-animal immunogenicity experiments. Importantly, the primary objective was to evaluate immunogenicity and functional antibody responses rather than protective efficacy. Building on these results, larger challenge and field trials are being planned to rigorously assess the vaccine's impact on EHEC colonization and shedding under production conditions.

Taken together, our results support EIT-PF as a broadly reactive proof-of-concept vaccine candidate for preharvest control of EHEC in cattle, while underscoring the need for larger, controlled challenge or field studies to assess its efficacy under production conditions.

## 5. Conclusions

In summary, this work establishes EIT-PF as a broadly reactive, low-cost proof-of-concept vaccine candidate against EHEC, although larger challenge and field studies will be essential to determine its protective efficacy in cattle.

## Supporting information

**S1 Fig. Effect of thermal shock on *E. coli* BL21 pTRC-EIT-K survival.** Bacterial pellets were resuspended in TE buffer and incubated at 55 °C. Every 10 min, aliquots were plated for CFU determination.
(TIF)

**S2 Fig. Analysis of plasmid stability.** After successive cultivation in antibiotic-free media, aliquots were plated on both LB with (black bars) and without (grey bars) kanamycin for CFU counting. Error bars correspond to standard deviations from replicates, data were analyzed by Student´s *t*-tests. ns: not significant.
(TIF)

**S3 Fig. Western-blot analysis of mice sera.** To test antibody specificity, purified MBP-EITH7 was run in SDS-PAGE and revealed with sera of 4 mice at 4 weeks post-immunization (M1 to M4) and a control mouse (CM) as primary antibodies.
(TIF)

**S4 Fig. EIT-specific total IgG antibody titers in steer sera.** Endpoint titers were determined for sera from control and EIT-immunized steers at the different post immunization days. ELISA endpoint titers are expressed as the base 2 logarithm of the reciprocal of the maximum dilution that resulted in an Abs 450nm value above the cut-off. The cut-off was defined as the average + 3SD of the levels measured for pre-immune sera. Each dot represents individual immunized steers (EIT-PF), or the control steer (C-PF) determined in triplicate, and the averages (±SEM) are shown for all data. PRE-I corresponds to pre-immune sera; C-PF I, II, III and EIT-PF I, II and III correspond to sera from steers immunized with control periplasmic fraction or EIT periplasmic fraction at week 2 (I dose), 4 (II doses) and 6 (III doses), respectively. Data were analyzed by One-way ANOVA and Sidak's post hoc contrasts; asterisks indicate a p-value < 0.0001 (****).
(TIF)

**S5 Fig. Pathogen-specific IgG reactivity.** Cross-reactivity of IgG antibodies was tested by coating ELISA plates with lysates of individual pathogen serotypes. Sera from one-dose EIT-PF immunized steers (week 2 p.i.) were pooled and assayed in triplicate, and the averages (±SEM) are shown. Data were analyzed by One-way ANOVA and Dunnett contrasts comparing to BL21 strain. Asterisks indicate a p-value < 0.001 (***) or < 0.0001 (****).
(TIF)

**S6 Fig. Antigen-specific IgG reactivity.** Reactivity of IgG antibodies was tested by coating ELISA plates with individual purified antigens, in addition to BSA as negative control. Sera from pre-immune and one-dose EIT-PF immunized steers (week 2 p.i.) were pooled and assayed in triplicate, and the averages (±SEM) are shown. Data were analyzed by One-way ANOVA and Sidak's post hoc contrasts; asterisks indicate a p-value < 0.05 (*) or < 0.0001 (****).
(TIF)

**S1 Raw Data. Raw numerical datasets and uncropped image files.** Raw numerical data and uncropped images used to generate all main and supplementary figures are available at the Figshare repository: DOI: 10.6084/m9.figshare.30764966.
(ZIP)

## Acknowledgments

PL is a graduate student at UNSAM, PF is UNSAM staff, and C.M.D. is affiliated with CONICET. L.A.B., D.G.N., M.S.R., and G.B. are members of the Research Career of CONICET and also serve as faculty members at UNSAM.

## Author contributions

**Conceptualization:** Laura A Basile, Mara S Roset, Gabriel Briones.

**Data curation:** Cecilia M. Duarte, Laura A Basile.

**Formal analysis:** Diego G Noseda, Mara S Roset, Gabriel Briones.

**Funding acquisition:** Gabriel Briones.

**Investigation:** Cecilia M. Duarte, Laura A Basile, Diego G Noseda, Pilar Leguiza, Gabriel Briones.

**Methodology:** Cecilia M. Duarte, Laura A Basile, Diego G Noseda, Pierina Fasolo.

**Supervision:** Gabriel Briones.

**Writing – original draft:** Laura A Basile, Gabriel Briones.

**Writing – review & editing:** Laura A Basile, Diego G Noseda, Mara S Roset, Gabriel Briones.

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
