## [Decision Letter · Decision Letter 0]

8 Oct 2025

Dear Dr. Briones,

We look forward to receiving your revised manuscript.

Kind regards,

Yung-Fu Chang

Academic Editor

PLOS ONE

Journal Requirements:

3. To comply with PLOS One submissions requirements, in your Methods section, please provide additional information regarding the experiments involving animals and ensure you have included details on (1) methods of sacrifice, (2) methods of anesthesia and/or analgesia, and (3) efforts to alleviate suffering.

5. Please expand the acronym “ANPCyT” (as indicated in your financial disclosure) so that it states the name of your funders in full.

This work was supported by grants from the ANPCyT of Argentina (PICT-2021-CAT-I-00049). PL is a graduate student at UNSAM, PF is UNSAM staff, and C.M.D. is a doctoral fellow from CONICET. L.A.B, D.G.N, M.S.R., and G.B are members of the Research Career of CONICET.

PICT-2021-CAT-I-00049. (ANPCyT)

7. Your ethics statement should only appear in the Methods section of your manuscript. If your ethics statement is written in any section besides the Methods, please move it to the Methods section and delete it from any other section. Please ensure that your ethics statement is included in your manuscript, as the ethics statement entered into the online submission form will not be published alongside your manuscript.

8. Please provide a complete Data Availability Statement in the submission form, ensuring you include all necessary access information or a reason for why you are unable to make your data freely accessible. If your research concerns only data provided within your submission, please write "All data are in the manuscript and/or supporting information files" as your Data Availability Statement.

Reviewers' comments:

Reviewer's Responses to Questions

**Comments to the Author**

1. Is the manuscript technically sound, and do the data support the conclusions?

Reviewer #1: Partly

Reviewer #2: Yes

2. Has the statistical analysis been performed appropriately and rigorously?

Reviewer #1: I Don't Know

Reviewer #2: Yes

3. Have the authors made all data underlying the findings in their manuscript fully available?

Reviewer #1: Yes

Reviewer #2: Yes

4. Is the manuscript presented in an intelligible fashion and written in standard English?

Reviewer #1: Yes

Reviewer #2: Yes

Reviewer #1: Review comment for PONE-D-25-51993

Duarte et al. reported that a chimeric fusion protein carrying peptides of EspA, intimin, and Tir of enterohemorrhagic E. coli induced IgG antibodies in the intraperitoneally immunized mice or subcutaneously immunized cattle, and the induced antibodies were functional against pedestal formation in vitro and reduced EHEC bacterial shedding in immunized mice after challenge with STEC O157, and further suggested that this protein extracted from periplasm with a low-cost thermal shock method can be for a cost-effective vaccine. While the study findings are interesting and indicate the potential of this vaccine candidate, additional experiments involving cattle challenges and efficacy assessments are needed.

Major:

1. While immunogenicity data, in vitro assays, and mouse infections against bacteria shedding indicate the potential of this vaccine candidate, efficacy from the cattle challenge study, even a small-scale pilot study, will be critical in providing direct evidence for assessing the vaccine's efficacy. As indicated by the authors ‘ further studies are needed to 43 assess its efficacy in controlling EHEC under field condition’, a cattle challenge study can significantly improve the quality of this manuscript.

2. Mouse immunization showed immune responses increased after boosters, especially for EIT-PF. The immunization in steer, however, showed no booster effect, and steer responded well after the prime. This often suggests pre-exposure. While the Abs from sera of pre-immunization were significantly lower, it cannot eliminate the chance of pre-exposure. Authors need to elaborate on it in the discussion.

3. Why no IgA responses in the immunized mice or steer?

4. Any data to show antibody functions directly against the binding between intimin and Tir?

Minor:

1. The figure resolution is poor and needs to be improved.

2. A brief description of ELISA coating antigen is needed, though references were cited. Were the antibodies derived from EIT-PF specific to EspA, intimin, and Tir? Using an ELISA coating antigen that is the same or similar to the immunogen is not the proper way to titrate antigen-specific antibody responses.

3. Change ‘neutralizing’ to ‘functional’.

4. Line 201-202: why PBS for the control group, not C-PF?

5. Figure 3B: the antibody response levels are the same for EIT-PF with or without an adjuvant after two boosters, why do antibody activities against pedestal formation differ significantly? Discussion on the function of the adjuvant is needed.

Reviewer #2: This manuscript describes the development of a low-cost vaccine candidate targeting enterohemorrhagic Escherichia coli (EHEC) for cattle, based on a chimeric antigen (EIT: EspA–Intimin–Tir) produced in E. coli BL21 and extracted through a simplified periplasmic protocol. The study integrates molecular design, small-animal validation, and preliminary immunization in steers, aiming to establish a scalable, antibiotic-free vaccine production platform. However, the current version presents limitations that preclude acceptance in its present form. The cattle trial includes too few animals to substantiate efficacy claims; antigen purity and quantitative characterization are incomplete; The manuscript would benefit from expanded discussion of limitations and a more balanced conclusion that frames the work as proof-of-concept rather than a validated vaccine. Below are the comments to improve the manuscript.

1. Cattle Trial Scope and Interpretation: Only six vaccinated and one control steer were tested. This limited sample size prevents meaningful statistical inference or generalization. The claim that a single dose may suffice should be presented as preliminary. Clarify the exploratory nature of the trial and temper conclusions regarding efficacy. Consider adding or referencing ongoing field or challenge trials.

2. Antigen Purity and Characterization: The “periplasmic fraction” likely contains multiple E. coli proteins. Contaminant proteins may contribute to immunogenicity. Provide SDS-PAGE or densitometric purity analysis or explicitly discuss this limitation in the Discussion.

3. Antibody Response Analysis: ELISA data show general reactivity but lack endpoint titers, isotype profiling, and cross-reactivity testing with other STEC serotypes. Functional assays (pedestal inhibition) are qualitative. Include quantitative neutralization indices or antibody titration curves, and if possible, analyze IgG subclasses or mucosal IgA.

4. The discussion highlights cost advantages but underrepresents biological and translational challenges (antigen stability, large-scale formulation, field efficacy). Add a paragraph acknowledging these limitations and outline next experimental steps.

5. Phrases such as “cost-effective vaccine candidate for cattle” should be qualified as preliminary or proof-of-concept, since protective efficacy in the target species is not yet demonstrated. Please add the conclusion to the manuscript.

**Do you want your identity to be public for this peer review?** For information about this choice, including consent withdrawal, please see our Privacy Policy

Reviewer #1: No

Reviewer #2: No

---

## [Author Response · Author response to Decision Letter 1]

3 Dec 2025

RESPONSE TO REVIEWERS (NOV-2025)

1. “Please ensure that your manuscript meets PLOS ONE's style requirements, including those for file naming”.

We have revised the manuscript to ensure full compliance with PLOS ONE’s formatting and style requirements. All sections, headings, and file names have been updated following the journal’s style templates available at the links provided. The manuscript now adheres to PLOS ONE’s guidelines for structure, formatting, and file naming conventions

2. “PLOS ONE now requires that authors provide the original uncropped and unadjusted images underlying all blot or gel results reported in a submission’s figures or Supporting Information files”. Blot gels/ figures/ This policy and the journal’s other requirements for blot/gel reporting and figure preparation are described in detail at https://journals.plos.org/plosone/s/figures#loc-blot-and-gel-reporting-requirements and https://journals.plos.org/plosone/s/figures#loc-preparing-figures-from-image-files. When you submit your revised manuscript, please ensure that your figures adhere fully to these guidelines and provide the original underlying images for all blot or gel data reported in your submission. See the following link for instructions on providing the original image data: https://journals.plos.org/plosone/s/figures#loc-original-images-for-blots-and-gels.

We have carefully reviewed the PLOS ONE guidelines for blot and gel reporting and have ensured that all figures in the revised manuscript fully adhere to these requirements. In accordance with the journal’s policy, we now provide all original, uncropped, and unadjusted blot and gel images underlying the results reported in the manuscript. These data have been compiled into a single Supporting Information file titled S1_Raw_Images. All raw images corresponding to the presented figures are available, and no underlying blot or gel data are missing.

3. To comply with PLOS One submissions requirements, in your Methods section, please provide additional information regarding the experiments involving animals and ensure you have included details on (1) methods of sacrifice, (2) methods of anesthesia and/or analgesia, and (3) efforts to alleviate suffering.

7. Your ethics statement should only appear in the Methods section of your manuscript.

Response to Editor Comment 3 and 7:

We have revised the Methods section to provide the additional animal welfare information requested by the journal. The Level 2 heading ‘Ethics statement’ now includes ethical approval, daily monitoring, and euthanasia procedures for mice, as well as vaccination and blood collection procedures for cattle. The ethics statement has been moved exclusively to this section, and all mentions of ethical approval in other sections of the manuscript have been removed, ensuring full compliance with the journal’s requirements.

The other Level 2 heading, ‘Vaccination experiments’ (formerly ‘Vaccination protocol’), contains the experimental procedures, with three Level 3 subheadings: ‘Mouse assays,’ ‘Steer assays,’ and ‘Challenge and fecal shedding of vaccinated mice.’ This reorganization separates ethical considerations from experimental procedures, ensures that all animal welfare aspects are clearly addressed, and clarifies the structure of the Methods section.

Specifically, the following text has been added in lanes 151-160:

Ethics statement

All animal studies were reviewed and approved by the Committee on the Ethics of Animal Experiments of the Universidad Nacional de San Martín (UNSAM) (permit number CICUAE UNSAM 15/2018), in accordance with international guidelines. Mice were monitored daily for signs of stress or discomfort and were euthanized by CO₂ inhalation at the end of the study, following institutional guidelines for humane endpoints. Cattle were immunized according to the vaccination schedule, and blood samples were collected at defined time points by trained personnel, with all efforts made to minimize stress or discomfort. No additional invasive procedures or euthanasia were performed in cattle.

4. We note that the grant information you provided in the ‘Funding Information’ and ‘Financial Disclosure’ sections do not match. When you resubmit, please ensure that you provide the correct grant numbers for the awards you received for your study in the ‘Funding Information’ section.

5. Please expand the acronym “ANPCyT” (as indicated in your financial disclosure) so that it states the name of your funders in full. This information should be included in your cover letter; we will change the online submission form on your behalf.

This work was supported by grants from the ANPCyT of Argentina (PICT-2021-CAT-I-00049). PL is a graduate student at UNSAM, PF is UNSAM staff, and C.M.D. is a doctoral fellow from CONICET. L.A.B, D.G.N, M.S.R., and G.B are members of the Research Career of CONICET.

We have carefully reviewed and updated the Funding Information and Financial Disclosure to ensure consistency. The acronym ‘ANPCyT’ has been expanded to its full name, ‘Agencia Nacional de Promoción Científica y Tecnológica (ANPCyT) of Argentina.’ All funding information has been moved exclusively to the Funding Statement in the online submission form, and any funding references have been removed from the Acknowledgments section. The Funding Statement now reads: ‘This work was supported by grants PICT-2021-CAT-I-00049 from ANPCyT (Agencia Nacional de Promoción Científica y Tecnológica of Argentina) and PIP 2021-2023 -11220200102517CO, P-UE Nº0086 from CONICET (Consejo Nacional de Investigaciones Científicas y Técnicas). This information was provided within the cover letter.

These revisions ensure full transparency and compliance with the journal’s requirements.

8. Please provide a complete Data Availability Statement in the submission form, ensuring you include all necessary access information or a reason for why you are unable to make your data freely accessible. If your research concerns only data provided within your submission, please write "All data are in the manuscript and/or supporting information files" as your Data Availability Statement.

We have provided a complete Data Availability Statement in the submission form. All data supporting the findings of this study are included in the manuscript and/or in the Supporting Information files, including the raw blot and gel images, in compliance with PLOS ONE’s guidelines.

Reviewer #1: Review comment for PONE-D-25-51993

Reviewer 1 – Major comment 1 (cattle challenge study)

Comment 1: While immunogenicity data, in vitro assays, and mouse infections against bacterial shedding indicate the potential of this vaccine candidate, efficacy from the cattle challenge study, even a small-scale pilot study, will be critical in providing direct evidence for assessing the vaccine's efficacy. As indicated by the authors ‘further studies are needed to assess its efficacy in controlling EHEC under field condition’, a cattle challenge study can significantly improve the quality of this manuscript.

We thank the reviewer for this important point and fully agree that a controlled cattle challenge would provide the most direct evidence of protective efficacy in the target species.

Practical and regulatory constraints prevented the inclusion of a cattle challenge within the present study. Performing EHEC challenge experiments in bovines requires specialized, high-biocontainment facilities, dedicated large-animal housing, and additional regulatory approvals that are not available for this initial, feasibility-oriented work. For these reasons, we designed (as suggested by both reviewers) the steer experiment as a preliminary, proof-of-concept assessment of immunogenicity rather than an efficacy trial. Although a formal cattle challenge was not performed here, we have strengthened the manuscript to make the limitation explicit and to present the available evidence that supports progression to challenge studies:

• We now clearly state throughout the manuscript that this is a proof-of-concept study and have tempered language implying demonstrated efficacy in cattle (Title, Abstract, Results, and Discussion).

• We report full in vivo protection data in the mouse model (challenge and accelerated clearance; Fig. 4), demonstrating that EIT-PF elicits protective responses in a controlled small-animal model.

• We present extensive bovine immunogenicity and functional data showing that sera from immunized steers: (i) develop robust antibody titers after a single EIT-PF dose (new figure S4), (ii) recognize multiple EHEC serotypes and EPEC (new figure S5), (iii) bind each component of the EIT chimera (EspA, Intimin, Tir; new figure S6), and (iv) exhibit functional activity in vitro by strongly inhibiting T3SS-dependent actin pedestal formation (Fig. 6). These functional in vitro data provide mechanistic evidence that the bovine antibody response targets critical virulence functions required for intestinal attachment.

To address the reviewer’s suggestion explicitly, we have added to the Discussion a paragraph that: (a) acknowledges the absence of a cattle challenge as a primary limitation, (b) describes the rationale for the periplasmic, low-cost antigen approach used here, and (c) outlines our planned next steps to perform controlled bovine challenge and field studies (Lines 440-467). These planned studies will be conducted only after securing the required biosafety approvals and access to suitable facilities.

We hope this combination of clear acknowledgment of the current limitation, presentation of robust supporting data (mouse protection & bovine functional sera), and a plan for future cattle challenge studies satisfactorily addresses the reviewer’s concern. Specific manuscript changes are noted in the revised file: Abstract (proof-of-concept language), Results (bovine immunogenicity and functional data — Figs. 5–6; S4–S6), and Discussion (new paragraph acknowledging limitation and outlining challenge plans).

2. Mouse immunization showed immune responses increased after boosters, especially for EIT-PF. The immunization in steer, however, showed no booster effect, and steer responded well after the prime. This often suggests pre-exposure. While the Abs from sera of pre-immunization were significantly lower, it cannot eliminate the chance of pre-exposure. Authors need to elaborate on it in the discussion.

We thank the reviewer for this insightful observation. We agree that the absence of a detectable booster effect in steers contrasts with the response observed in mice and may raise the possibility of prior exposure to EHEC or related antigens. In response to this comment, we have expanded the Discussion to address this point in detail.

Specifically, we now clarify that although pre-immunization sera showed only background reactivity in both ELISA and functional assays, suggesting no ongoing or recent infection, we cannot completely exclude the possibility of previous natural exposure, as transient colonization with EHEC or related E. coli strains is common in cattle and may prime the immune system without leaving persistent serological signatures.

This clarification has been incorporated into the revised Discussion (Lines 452-459).

3. Why no IgA responses in the immunized mice or steer?

We appreciate the reviewer’s observation. As our vaccine formulation was administered parenterally, rather than via mucosal routes, the induction of antigen-specific IgA was not expected. Systemic vaccination predominantly elicits IgG responses, while robust IgA production generally requires mucosal delivery strategies (oral, intranasal, or rectal), often combined with specific mucosal adjuvants.

4. Any data to show antibody functions directly against the binding between intimin and Tir?

We appreciate the reviewer’s insightful question. Although we do not yet have a purified biochemical assay that directly measures the blockade of the Intimin–Tir interaction, we performed two complementary analyses that address this point both antigen-specifically and functionally. First, in the new Figure S6, ELISA experiments included individually purified recombinant EspA, Intimin, and Tir domains, with the latter two containing the defined interaction regions involved in Tir–Intimin binding. This approach allowed us to detect antibodies recognizing each component independently, including those targeting epitopes within the Tir–Intimin interface. Second, to assess functional relevance, we used the pedestal-formation inhibition assay, which integrates the entire T3SS-dependent sequence of events—Tir translocation, Tir–Intimin engagement, and actin pedestal assembly. Although indirect, this assay is widely accepted as a functional surrogate for interference with the Tir–Intimin pathway. Taken together, these results demonstrate that vaccinated steers generate antibodies against each domain of the chimera, including the key interaction regions, and that these antibodies are functionally capable of interfering with T3SS-driven pedestal formation.

Minor:

1. The figure resolution is poor and needs to be improved.

We appreciate the comment. Main figures have now been replaced with high-resolution versions and all figures have been prepared following the journal’s guidelines and checked through PACE tool to ensure clarity and readability.

2. A brief description of ELISA coating antigen is needed, though references were cited. Were the antibodies derived from EIT-PF specific to EspA, intimin, and Tir? Using an ELISA coating antigen that is the same or similar to the immunogen is not the proper way to titrate antigen-specific antibody responses.

A brief description of the antigen used to coat ELISA plates has been added (new lines 193–194), together with an explanation of the individual purified domains employed to assess EspA, Intimin, and Tir specificity (new lines 217–221). We agree that coating ELISA plates with an antigen identical to the immunogen may not fully reflect antigen-specific titers. However, in our study, the immunogen was the whole periplasmic fraction, whereas ELISA plates were coated with affinity-purified recombinant EIT, enabling selective detection of EIT-specific antibodies. Importantly, control steers immunized with periplasmic extracts lacking EIT showed no detectable signal, confirming that the assay does not measure background reactivity to E. coli periplasmic proteins.

Additionally, we now provide a new figure showing ELISA assays against the individually purified EspA, Intimin, and Tir domains (the latter two corresponding to their interaction regions, as detailed in Methods) (Figure S6). These data demonstrate that bovine sera recognize all three components of the chimera, supporting the specificity of the antibody response elicited by EIT-PF.

3. Change ‘neutralizing’ to ‘functional’.

Done. All mentions of “neutralizing antibodies” have been replaced with “functional antibodies” to more accurately reflect the biological activity measured.

4. Line 201-202: why PBS for the control group, not C-PF?

This was a mistake introduced during manuscript editing. It has been corrected in the revised version (lines 183 and 350).

5. Figure 3B: the antibody response levels are the same for EIT-PF with or without an adjuvant after two boosters, why do antibody activities against pedestal formation differ significantly? Discussion on the function of the adjuvant is needed.

We thank the reviewer for this comment. While ELISA signals were similar between adjuvanted and non-adjuvanted groups, inclusion of the aluminum-based adjuvant enhanced functional activity, improving antibodies’ ability to block pedestal formation. This is consistent with reports that alum can promote higher-quality, functional antibodies beyond simply increasing titers (Engineered immunogen binding to alum adjuvant enhances humoral immunity, PMC706

---

## [Decision Letter · Decision Letter 1]

2 Jan 2026

Developing a simple, cost-effective proof-of-concept vaccine candidate against EHEC for cattle

PONE-D-25-51993R1

Dear Dr.  Briones,

We’re pleased to inform you that your manuscript has been judged scientifically suitable for publication and will be formally accepted for publication once it meets all outstanding technical requirements.

Kind regards,

Yung-Fu Chang

Academic Editor

PLOS One

Additional Editor Comments (optional):

Reviewers' comments:

Reviewer's Responses to Questions

**Comments to the Author**

Reviewer #1: All comments have been addressed

Reviewer #2: All comments have been addressed

2. Is the manuscript technically sound, and do the data support the conclusions?

Reviewer #1: Yes

Reviewer #2: Yes

3. Has the statistical analysis been performed appropriately and rigorously?

Reviewer #1: I Don't Know

Reviewer #2: Yes

4. Have the authors made all data underlying the findings in their manuscript fully available?

Reviewer #1: Yes

Reviewer #2: Yes

5. Is the manuscript presented in an intelligible fashion and written in standard English?

Reviewer #1: Yes

Reviewer #2: Yes

Reviewer #1: (No Response)

Reviewer #2: I have carefully reviewed the revised manuscript in full and confirm that all reviewer comments have been thoroughly addressed. The manuscript has been clearly framed as a proof-of-concept study, with claims appropriately tempered throughout the research article. I believe the manuscript now meets the scientific, ethical, and reporting standards of PLOS ONE and is suitable for final editorial consideration.

**Do you want your identity to be public for this peer review?** For information about this choice, including consent withdrawal, please see our Privacy Policy

Reviewer #1: No

Reviewer #2: **Yes:**  Mohd Abdullah

---

## [Editor Report · Acceptance letter]

PONE-D-25-51993R1

PLOS One

Dear Dr. Briones,

I'm pleased to inform you that your manuscript has been deemed suitable for publication in PLOS One. Congratulations! Your manuscript is now being handed over to our production team.

Kind regards,

on behalf of

Dr. Yung-Fu Chang

Academic Editor

PLOS One